# Electrocatalytic CO_2_ Reduction and H_2_ Evolution by a Copper (II) Complex with Redox-Active Ligand

**DOI:** 10.3390/molecules27041399

**Published:** 2022-02-18

**Authors:** Jingjing Li, Shifu Zhang, Jinmiao Wang, Xiaomeng Yin, Zhenxing Han, Guobo Chen, Dongmei Zhang, Mei Wang

**Affiliations:** 1Key Laboratory of Marine Chemistry Theory and Technology, Ministry of Education, College of Chemistry and Chemical Engineering, Ocean University of China, Qingdao 266100, China; ljj11@stu.ouc.edu.cn (J.L.); zhangshifu@stu.ouc.edu.cn (S.Z.); wangjinmiao@stu.ouc.edu.cn (J.W.); yinxiaomeng@stu.ouc.edu.cn (X.Y.); hanzhenxing66@163.com (Z.H.); chenguobo@ouc.edu.cn (G.C.); zdmei@ouc.edu.cn (D.Z.); 2Institute for New Energy Materials & Low Carbon Technologies, School of Materials Science and Engineering, Tianjin University of Technology, Tianjin 300384, China

**Keywords:** copper (II) complex, redox-active ligand, electrocatalysis, CO_2_ reduction, H_2_ evolution

## Abstract

The process of electrocatalytic CO_2_ reduction and H_2_ evolution from water, regarding renewable energy, has become one of the global solutions to problems related to energy consumption and environmental degradation. In order to promote the electrocatalytic reactivity, the study of the role of ligands in catalysis has attracted more and more attention. Herein, we have developed a copper (II) complex with redox-active ligand [Cu(**L^1^**)_2_NO_3_]NO_3_ (**1**, **L^1^** = 2-(6-methoxypyridin-2-yl)-6-nitro-1h-benzo [D] imidazole). X-ray crystallography reveals that the Cu ion in cation of complex **1** is coordinated by two redox ligands **L^1^** and one labile nitrate ligand, which could assist the metal center for catalysis. The longer Cu-O bond between the metal center and the labile nitrate ligand would break to provide an open coordination site for the binding of the substrate during the catalytic process. The electrocatalytic investigation combined with DFT calculations demonstrate that the copper (II) complex could homogeneously catalyze CO_2_ reduction towards CO and H_2_ evolution, and this could occur with great performance due to the cooperative effect between the central Cu (II) ion and the redox- active ligand **L^1^**. Further, we discovered that the added proton source H_2_O and TsOH·H_2_O (p-Toluenesulfonic acid) could greatly enhance its electrocatalytic activity for CO_2_ reduction and H_2_ evolution, respectively.

## 1. Introduction

The enormous global consumption of fossil fuel leads to critical environmental, energy, and climate issues [1]. Several promising strategies have been developed to resolve the above issues. Firstly, applying renewable electricity to drive CO_2_ into high-value fuels such as carbon monoxide (CO), formic acid (HCOOH), methanol (CH_3_OH), and methane (CH_4_), etc., has been considered a sustainable route to alleviate energy shortages and global warming [2]. Secondly, as a green energy and high energy carrier, hydrogen (H_2_) is a great candidate to replace fossil fuels in the near future [3]. Therefore, catalytic hydrogen evolution reaction (HER), with high efficiency and low cost, has become a hot research area [4]. Thirdly, due to the similar reduction potentials, HER is the main competing reaction during electrocatalytic CO_2_ reduction reaction (CO_2_RR), which is a crucial problem to be solved for CO_2_RR [5]. Nevertheless, the competition reactions of HER and CO_2_RR can be rationally used in exploring efficient ways to produce syngas, the gaseous mixture of CO and H_2_, which are very useful in producing fuels such as alcohols, hydrocarbon, dimethyl ether, and more fuels through the industrial Fischer–Tropsch process, etc. [6].

In recent years, the uses of low-cost non-precious transition metal complexes as catalysts have aroused the interest of researchers. Among these non-precious metal complexes, due to the special electron structure of copper, copper complexes have been confirmed to be effective electrocatalysts in various reactions. In 2021, Lan et al. reported a copper (I) based complex with excellent electrocatalytic selectivity for CO_2_-to-CH_4_ reduction, which occurs because of the chalcophile interactions in crystalline catalysts [7]. Mitsopoulou and coworkers investigated the electrocatalytic activity for hydrogen evolution of a Cu (I) diimine complex, and they found that the coordinated nitrogen atoms play an important role, while Yang’s group developed a copper (0) enriched material to tune the syngas production with great electrochemical activity [8,9].

In order to promote the electrocatalytic reactivity, the study of the role of ligands in catalysis has attracted more and more attention [10]. Redox-active ligands, e.g. non-innocent ligands, have been affirmed to be a multifunctional tool for improving electrocatalysis both dynamically and thermodynamically, because they can act as an electron reservoir to accept or donate electrons, regulating the electronic properties of the central metal ions, etc. [11,12,13]. Those extraordinary features enable the redox active ligands to facilitate various catalysis of the transition metal complexes with high reactivity. Jurss’ group reported a cobalt (II) complex with bipyridyl-NHC ligand, showing high selectivity for electrocatalytic CO_2_ reduction to CO, occurring due to the redox-active synergy effect between the cobalt center and redox-active ligands [14]. Our group also explored the crucial effect of the redox-active ligand bis(imino)pyridine (PDI) on the electrocatalytic activity for CO_2_ reduction [15]. Hess and co-workers investigated the electrocatalytic H_2_ evolution by a Co (II) complex bearing macrocycle ligand, and they found that the redox-active ligand could modulate the energy and activity of HER [16]. Marinescu’s group prepared a cobalt (II) complex with thiolate ligand, and they found that protonation of the redox non-innocent ligand could influence the electrocatalytic reactivity for syngas generation [17]. Imidazole and its derivatives, as forms of redox-active ligands with various structures and rich electrons, have been proven to assist and even stabilize the metal centers in electrocatalytic reactions [18,19,20].

Regarding the aforementioned reasons, we have synthesized an imidazole derivative redox-active ligand **L^1^**, 2-(6-methoxypyridin-2-yl)-6-nitro-1h-benzo [D] imidazole, and obtained a copper (II) complex [Cu(**L^1^**)_2_NO_3_]NO_3_ (**1**) in which the central Cu (II) ion is coordinated by this ligand. The cation of complex **1** is coordinated by two redox ligands **L^1^** and one labile nitrate ligand. The systematic electrocatalysis investigation of complex **1** reveals that it can homogeneously electrocatalyze CO_2_ reduction and H_2_ evolution, and this occurs with great performance owing to the cooperative effect between the redox-active ligand **L^1^** and the metal center Cu (II). The added proton source, H_2_O, could highly enhance its electrocatalytic efficiency for CO_2_ reduction to CO. Meanwhile, in the process of catalytic CO_2_ reduction to CO, there is an inevitable competition reaction of hydrogen generation, which is a promising process for the synthesis of syngas, with the mixture of CO and H_2_. Therefore, we also investigated its catalytic reactivity for H_2_ evolution, and we found that the added proton source TsOH·H_2_O could highly enhance its electrocatalytic activity for H_2_ evolution.

## 2. Results and Discussion

### 2.1. The Nature and Character of the Complex

X-ray crystallography demonstrates that complex **1** belongs to monoclinic crystal and the space group is *C12/c1* system (Appendix A). The cation of complex **1** is surrounded by two redox-active ligands **L^1^** and one labile nitrate ligand. According to the bond valence sum calculations, the oxidation state of the copper atom in the complex is +2 [21]. As presented in Appendix A, the magnetic moment (μeff) of complex **1** at room temperature is around 1.80 µB, and, in addition, the ligand **L^1^** and the copper ion did not undergo reduction during the synthesis procedure, which confirms that the copper ion should be in +2 oxidation state. Furthermore, according to the bond valence sum calculations, the copper atom in the complex is in +2 oxidation state [21]. The metal center Cu^II^ in the complex is coordinated with four N atoms from the ligand **L^1^** and two O atoms from the nitrate ligand, forming a twisted tetravacant octahedral construction, as illustrated in Figure 1 and Appendix A (according to the analysis of SHAPE). The coordination geometry of complex **1** can be classified as a distorted four vacancy octahedron (vOC-2, 3C2v). Due to the special d^9^ electronic configuration of the metal center Cu^II^ ion, it is witnessed that the Jahn-Teller effect and Bailar distortions of the octahedron and Cu1-N6 and Cu1-O4 bonds are distorted almost on the same axis [22,23,24]. Meanwhile, the longer Cu-O bond between the metal center and the labile nitrate ligand would break to provide an open coordination site for the binding of the substrate during the catalytic process [25,26].

### 2.2. Electrochemistry under Atmosphere of 1 atm Ar

The cyclic voltammograms (CVs) of complex **1**, obtained at different scanning rates (100–500 mV s^−1^), in the electrolyte solution of 0.1 M ^n^Bu_4_NPF_6_/CH_3_CN in the argon (Ar) atmosphere, are displayed in Figure 2. When scanning towards cathode potentials, it can be observed that the complex has three irreversible reduction peaks at the potentials of −1.38 V, −1.97 V and −2.59 V vs. NHE (all the potentials are versus NHE) (Figure 2). Based on the DFT calculation, the LUMOs (lowest unoccupied molecular orbital) of complex **1** are mainly localized on the redox-active ligand **L^1^**. (Figure 3, the frontier molecular orbital surfaces of **1** are depicted in Appendix A). We have performed the Cyclic voltammetry under Ar atmosphere for the ligand **L^1^** (Appendix A), which displays two reduction peaks at the potential of −0.48 V and −0.93 V. By comparing the reduction potentials of the ligand **L^1^**, the first two reduction peaks of complex **1** may be attributed to the reduction of the two redox active ligands **L^1^** and the according two radical anions [**L^1•^**]. Meanwhile, the last reduction wave can be ascribed to the Cu^II^/Cu^I^ couple. In addition, as shown in Figure 2b, the cathode current peaks (ip) at different scanning rates have a good linear correlation with the square root of the scanning rates, which proves that the electrode process is mainly a diffusion control process. We have listed all the data in these figures in the corresponding tables, as shown in Appendix A.

### 2.3. Electrochemistry in the Presence of CO_2_

The electrocatalytic performance for CO_2_ reduction of compound **1** was investigated under saturated CO_2_ atmosphere with 0.1 M ^n^Bu_4_NPF_6_ as the supporting electrolyte in MeCN solution. The cyclic voltammetry curves at −1.15 V under 1 atm CO_2_ are subtracted from the base line. We mainly studied the comparison of catalytic current of complex **1** at 100 mV s^−1^ in CO_2_ and Ar at −1.15 V, as shown in Appendix A. By carrying out cyclic voltammetry experiments on glassy carbon (GC) electrodes, we found that the CV plot in CO_2_ displays an enhanced irreversible reduction wave at −1.15 V compared with that in Ar atmosphere at the same scanning rate, while it also repeats very well at different scanning rates without new oxidation or reduction peaks emerging, as illustrated in Figure 4. In addition, we used FTO as the working electrode to perform controlled potential electrolysis of 2mM complex **1** in order to characterize the stability of our catalyst and determine the Faraday efficiency according to the literature reported before [27], and we detected CO based on GC (Gas Chromatography) analysis. All these results indicate that compound **1** can electrocatalytically reduce CO_2_ to CO with great stability. Additionally, as illustrated in Figure 5, in order to explore the electrocatalytic activity for CO_2_-to-CO of complex **1** in the presence of proton donor, different concentrations of H_2_O were added to the CO_2_ saturated MeCN solution as the proton source for the CV experiments under the same condition. We have observed that, by the addition of H_2_O, the reductive peak current density increases, which suggests that the addition of proton source can promote its catalytic reactivity for CO_2_ reduction, and the catalytic reaction should be a PCET process [25]. Furthermore, Figure 6 and Appendix A show the concentrations of complex **1** as exhibiting linear relationship with the catalytic currents at the catalytic potential of −1.15 V, which suggests that the catalytic CO_2_-to-CO conversion is the first-order reaction.

The turnaround frequency (TOF) of the electrocatalytic CO_2_ of complex **1** can be determined by the Equation (1) below:(1)$$\mathrm{TOF}=\frac{Fvn_{p}^{3}}{RT}{\left(\frac{0.4463}{n_{cat}}\right)}^{2}{\left(\frac{i_{cat}}{i_{p}}\right)}^{2}$$

*F* is the Faraday constant (96,485 C·mol^−1^), *v* is the scanning rate used (0.1 V s^−1^), *n*_p_ is the number of electrons involved in the non-catalytic oxidation reduction reaction (*n*_p_ = 1), and *R* is the gas constant (8.314 J·K^−1^·mol^−1^), *T* is temperature (293.15 K), *n*_cat_ is the number of electrons involved in the catalytic reaction (*n*_cat_ = 2 indicates the reduction of CO_2_ to carbon monoxide), *i*_p_ and *i*_cat_ are identified as peak currents under Ar and CO_2_, respectively. By Equation (1), TOF is calculated as 0.65 s^−1^ at the potential of −1.15 V vs. NHE (*i*_cat_/*i*_p_ = 1.82), which is comparable with those reported cooper based homogeneous catalysts [28,29,30,31].

In order to further explore the electrocatalytic ability of this complex for CO_2_ reduction, a series of CPE experiments were recorded in CO_2_ saturated CH_3_CN solution, with the addition of distilled water as the proton source. As shown in Appendix A, by the red line, during 4000 s electrolysis, the current density can reach ~−1.5 mA cm^−2^ at the potential of −1.15 V vs. NHE in present of 2 mM complex **1** and 0.139 mM water; the mixed-gas CO and H_2_ are detected, which reveals that complex **1** is a potential catalyst for producing syngas [27,28,29]. As demonstrated in Appendix A, according to the GC analysis, the calculated Faraday efficiency (FE) of CO evolution is nearly 10% and is about 90%. Furthermore, the current density is very small at the potential of −1.15 V vs. NHE without complex **1** (the black line), and no CO or H_2_ is detected, indicating that no catalysis occurs. Additionally, we also examined the solution after electrolysis by MS analysis, but we did not observe HCOOH or CH_3_OH. Moreover, the almost linear curve of CPE indicates that catalysts **1** can remain stable in solution throughout the catalytic process. Meanwhile, as shown in Appendix A, the rinse test was also conducted on the FTO glass electrode after electrocatalysis, displaying nearly no current density, similar with that of the blank test before the catalysis, which assures that complex **1** is a stable homogeneous catalyst. Additionally, we have carried out DLS of (dynamic light scatting) of complex **1** before and after 4000 s electrolysis in MeCN solution (Appendix A, which reveals that the particle distributions are in the range of the molecular hydrodynamic diameter of the cluster, indicating that there are no nanoparticles formed during electrolysis. In addition, Appendix A illustrates the in-situ UV–vis spectroelectrochemistry of complex **1**, conducted during 4000 s CPE, which shows that there is close to no difference in the spectrum, proving the high stability of catalyst **1** during electrolysis. During the reduction process, the longer Cu-O bond between the metal center and the labile nitrate ligand could break to provide an open coordination site for the binding of the substrate CO_2_ to produce the possible intermediate Cu-COO* [23,24]. More importantly, via two reduction steps, the two redox-active ligands are both reduced to radial two radical anions [**L^1•^**], which could assist the mental center to cooperatively catalyze CO_2_ reduction [10].

### 2.4. Electrocatalytic Property for Hydrogen Evolution

The electrocatalytic property for hydrogen evolution (HER) of complex **1** was also investigated in CH_3_CN (0.1 M ^n^Bu_4_NPF_6_) electrolyte solution with the addition of p-Toluenesulfonic acid (TsOH·H_2_O) as the proton source. As depicted in Figure 7, compared with the CV plot without addition of the proton source, it can be found that, with the increasing amounts of p-Toluenesulfonic acid added in the solution, the peak currents at the potential of −1.97 V are greatly enhanced. Meanwhile, we detected amounts of H_2_ by GC analysis, by CPE experiments, at this potential. These results prove that the reduction reaction of H^+^ can occur in the presence of complex **1** at the presence of p-Toluenesulfonic acid. In order to further study the electrocatalytic ability of complex **1** for hydrogen evolution, a series of CPE experiments were carried out for 4000 s in CH_3_CN with and without p-Toluenesulfonic acid at different potentials using FTO working electrode, which suggest that, with 0.58 mM of p-Toluenesulfonic acid at the potential of −1.97 V, the current density can reach ~5 mA cm^−2^, as illustrated in Figure 8. Furthermore, in order to prove that H_2_ is not produced by TsOH·H_2_O, the comparative CV experiment, containing solely TsOH·H_2_O in CH_3_CN solution without complex **1**, is carried out, indicating that there are no reduction waves around −1.97 V (Appendix A). Beyond this finding, in the absence of complex **1**, the current density at −1.97 V is negligible, which suggests that no catalytic reaction occurs without complex **1**. According to Equation (1), Appendix A, the calculated TOF of H_2_ evolution is 0.33 s^−1^, and FE is shown in Appendix A, which is comparable with those reported copper based homogeneous catalysts [8,32,33,34].

Moreover, the rinse test conducted on the FTO glass electrode after electrocatalysis shows almost no current density, which resembles that of the blank test before the catalysis (Figure 8), revealing that complex **1** can homogeneously catalyze H^+^ reduction with high stability. The DLS of complex **1** proved that no nanoparticles were formed before and after electrolysis for 4000 s, indicating that it has excellent stability, as illuminated in Appendix A. Additionally, the nearly linear curve of CPE indicates that catalyst **1** can remain stable in solution throughout the catalytic process. Furthermore, as shown in Appendix A, during CPE for 4000 s, the in-situ UV–vis spectroelectrochemistry of complex **1** displays negligible difference in the spectrum, confirming the great stability of catalyst **1** during electrolysis.

## 3. Materials and Methods

### 3.1. Synthesis

The ligand **L^1^** = 2-(6-methoxypyridin-2-yl)-6-nitro-1h-benzo [D] imidazole was synthesized according to the previous report [35]. Cu(NO_3_)_2_·3H_2_O (0.242 g, 1 mmol) was added to 15 mL acetonitrile solution of **L^1^** (0.782 g, 2 mmol), and stirred at room temperature for 12 h. The resulting solution was filtered, and the filtrate was kept for evaporation at room temperature for about 7 days to give X-ray-quality dark green crystals. Yield: 0.384 g (57.9%). Calc. (Found) for C_26_H_18_CuN_10_O_12_: C, 48.75(48.62); H, 2.81(2.99); N, 19.69(19.54). IR (KBr disk, cm^−1^): 3092 (m), 1591 (m), 1471 (m), 1422 (m), 1319 (s), 1047 (w), 803 (w), 732(w), and 607(w) (Appendix A. The purity of the synthesized complex **1** was confirmed by powder X-ray diffraction (PXRD) analysis, which shows that the peak positions of the diffraction in their experimental and theoretical PXRD patterns all agreed well (Appendix A, demonstrating that the prepared samples are all pure.

### 3.2. General Materials and Characterization

The solvents and materials used are reagent grade and have not been purified. Unless otherwise stated, all the operations are carried out under aerobic conditions, all the chemicals are commercially available and can be used without further purification, and the carbon dioxide and argon are purchased from Dehai Gas Company (Hainan, China). X-ray powder diffraction (XRD) (Appendix A) is carried out on the Bruker D8 powder diffraction instrument in order to obtain the purity of the complement and the sample of the complex. After complex fracks with pure KBr, infrared spectral data (Appendix A) is recorded by the Nicolet 170SX infrared spectrometer (Thermo Fisher, Waltham, MA, USA) in the 4000–500 cm^−1^ scanning range. Elemental analysis uses 2400 PerkinElmer analyzers to examine the percentage content of C, H, and N elements of the mates, and the theoretical values are basically consistent.

### 3.3. Crystal Structure Determination

The structural data of crystal is collected using the Bruker Smart-1000 CCD X-ray monocrystalline diffraction instrument (Bruker, Germany), all of which is restored by the SAINT v8.34A program (Bruker, 2013) and corrected and absorbed using the SADABS program (Bruker, 2014/5). The SHELXL software (v.2014/7) and Olex2 software (v.1.2) parse the initial structure by direct method, refining it with F^2^-based full matrix least square technology [36,37], and this technique is used to modify the non-O atomic coordinates and anisotropy. Appendix A gives cell parameters, spatial groups, some conventional thermodynamic parameters, and other data of the crystal, and it introduces the relevant crystal information in detail. Appendix A list the selected key length and key angles. The anion of the molecule is severely disordered, thus a Platon-Squeeze [38] was used to refine the anion-free structure of complex **1**.

### 3.4. Electrochemical Measurement and Electrolytic Product Analysis

All electrochemical experiments are tested with CHI660E electrochemical analyzers in order to study their electrocatalytic properties, and these experiments are conducted in single-chamber three-electrode reactors. A solution of 0.1 M ^n^Bu_4_NPF_6_ in a dry acetonitrile was used as the supporting electrolyte. Cyclic voltammogram (CV) experiments were carried out using a glass carbon working electrode with a diameter of 3 mm, which was carefully polished with diamond plaster, and ultrasonically cleaned in aqueous ethanol and deionized water, and then dried before use.

There is close to 20 mL of solution in the electrolytic cell, and the concentration of the complex in the solution is close to 0.1 mol/L. The anti-electrode is platinum wire, and the reference electrode is the Ag/AgCl electrode. A conductive glass substrate doped with fluorine tin oxide (FTO) (1 cm × 1 cm, effective surface area of 1.0 cm^2^) (produced by Zhuhai Kaivo Optoelectronic Corp., Zhuhai, China) is used as an operating electrode to control potential electrolysis (CPE), which is soaked with 5 wt% NaOH in ethanol solution for several hours, and then washed with water, ethanol, and water in turn. Before each experiment, the solution is blown away at room temperature with Ar or CO_2_ for 30 min. CPE at the same condition on GCE is shown in Appendix A.

In-situ UV-visible spectral electrochemistry is performed by applying the constant potentials of −1.15 V (Appendix A) and −1.97 V vs. NHE (Appendix A) under CO_2_ and Ar atmosphere, respectively, and recorded using the UV-visible spectrophometometer (Shimadzu, Kyoto, Japan). In-situ spectral electrochemistry studies use quartz dishes with a path length of 1 cm as electrochemical batteries, including platinum mesh, platinum wire, and Ag/AgCl (saturated KCl) electrodes, respectively, as working electrodes, anti-electrodes, and reference electrodes. The top space gas sample (2 mL) produced by the capillary tube electrophoresis experiment is extracted using a bait-locked airtight syringe and injected into the gas chromatography (GC, Shimadzu GC-2014), equipped with a flame ionization detector (FID) containing a mechanical device to analyze carbon monoxide, and equipped with a thermal conductivity detector (TCD, Shimadzu) for analysis to quantify H_2_. Detection of CO_2_ and H_2_ are carried out with ultra-high purity Ar as carrier gas. Liquid products are analyzed by NMR (Bruker AVANCE III HD).

### 3.5. Density Functional Theory Calculations

Quantum-mechanical calculations were carried out utilizing the Gaussian 09 program package, using the B3LYP hybrid functional [39,40]; the “double-ξ” quality LanL2DZ [41] basis sets were used for transition metals (Cu), and 6-311G (d, p) basis sets were used for non-metal atoms [42]. The atom coordinates used in the calculations were gained from crystallographic data, and a molecule in the unit cells was selected as the initial model.

## 4. Conclusions

In this work, we have successfully synthesized a novel copper (II) electrocatalyst [Cu(**L^1^**)_2_NO_3_]NO_3_ (**1**) containing the redox active ligand 2-(6-methyl-2-base)-6-nitro-1H-benzene and [d] imidazole (**L^1^**). Through investigation, we discovered that complex **1** can electrocatalyze CO_2_ reduction to CO and HER. While adding water as the proton source in the system for CO_2_ reduction, the reactivity for CO_2_ reduction of complex **1** is enhanced. However, due to the competitive reaction of H^+^ reduction, the mixed gas CO and H_2_ are both evolved with the FE of 10% and 90%, respectively. Resultingly, we additionally studied its catalytic activity for HER, and observed that, with the increasing amounts of p-Toluenesulfonic acid added in the solution, the electrocatalytic reactivity for HER increased. Furthermore, in-situ UV–vis spectroelectrochemistry of complex **1** during CO_2_ reduction and HER were both carried out, and these both displayed negligible difference in the spectrum, confirming the great stability of catalyst **1** during electrolysis. Combined with DFT calculation, it has been confirmed that the great electrocatalytic performance of complex **1** is owning to the synergistic effect between the metal center Cu (II) and the redox-active ligand **L^1^**.

## Figures and Tables

**Figure 1 molecules-27-01399-f001:**
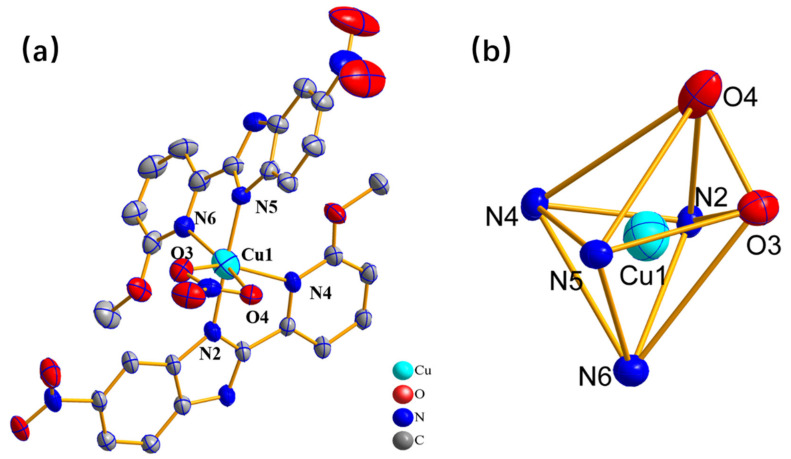
(**a**) The crystal structure of the complex **1**. (**b**) The spatial configuration of the complex **1**, all are 50% probability ellipses. Color codes: green, Cu; red, O; blue, N; gray, C; and hydrogen atoms are omitted for clear visibility.

**Figure 2 molecules-27-01399-f002:**
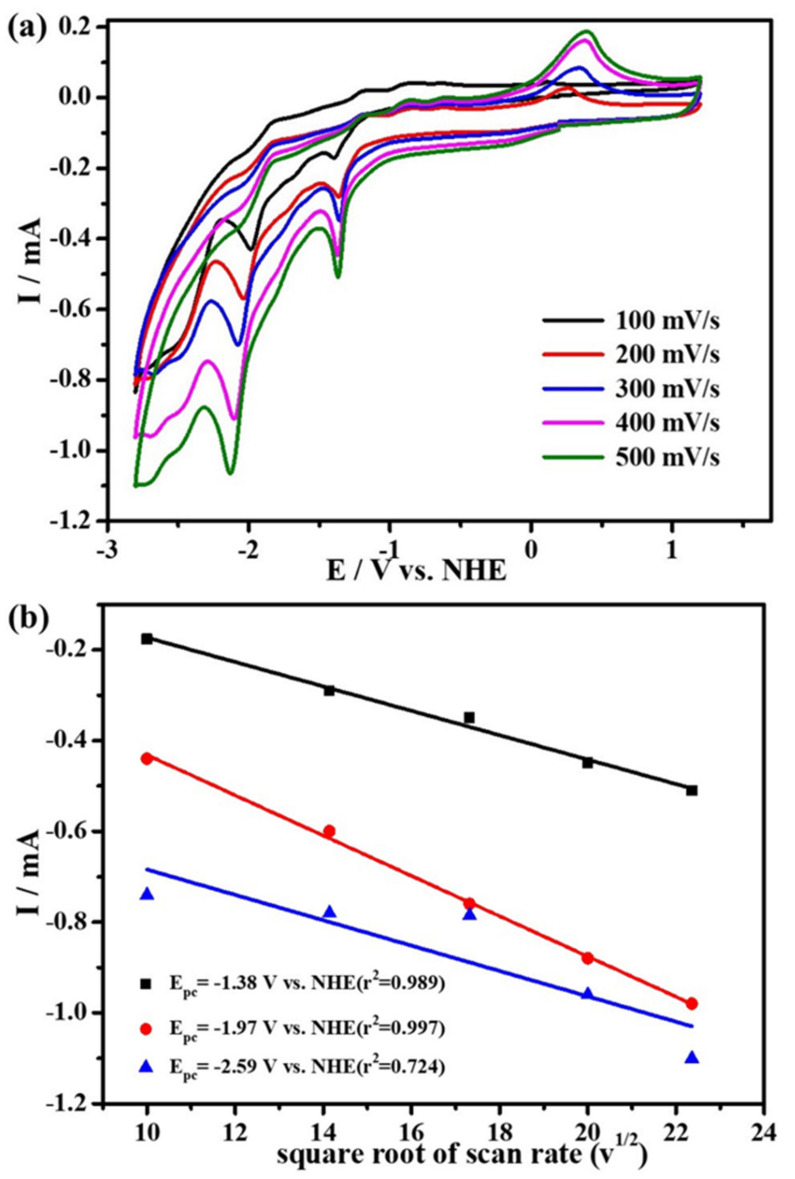
(**a**) cyclic voltammetry of 2 mM complex **1** under 1 atm Ar with 0.1 M ^n^Bu_4_NPF_6_ as supporting electrolyte at scan rates range from 100 to 500 mV s^−1^; and (**b**) the linear relationships between the peak cathodic currents and the square root of scan rates.

**Figure 3 molecules-27-01399-f003:**
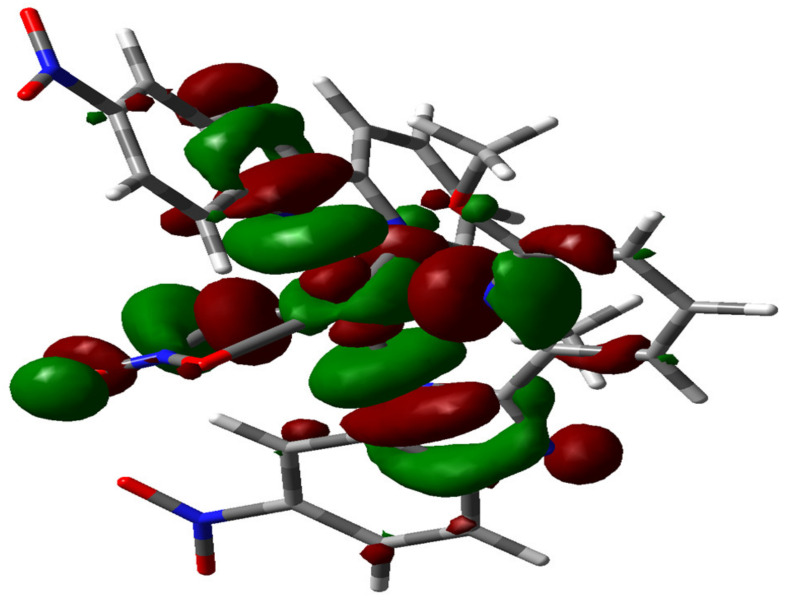
LUMO of complex **1** (iso value = 0.02).

**Figure 4 molecules-27-01399-f004:**
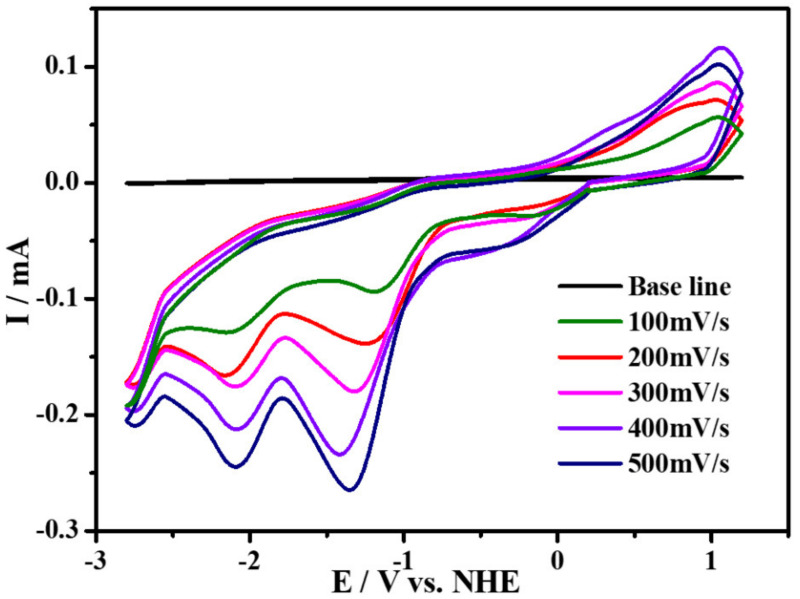
Cyclic voltammograms of 2 mM complex under 1 atm CO_2_ at scan rates range from 100 to 500 mV s^−1^ (all the plots are subtracted from the baseline).

**Figure 5 molecules-27-01399-f005:**
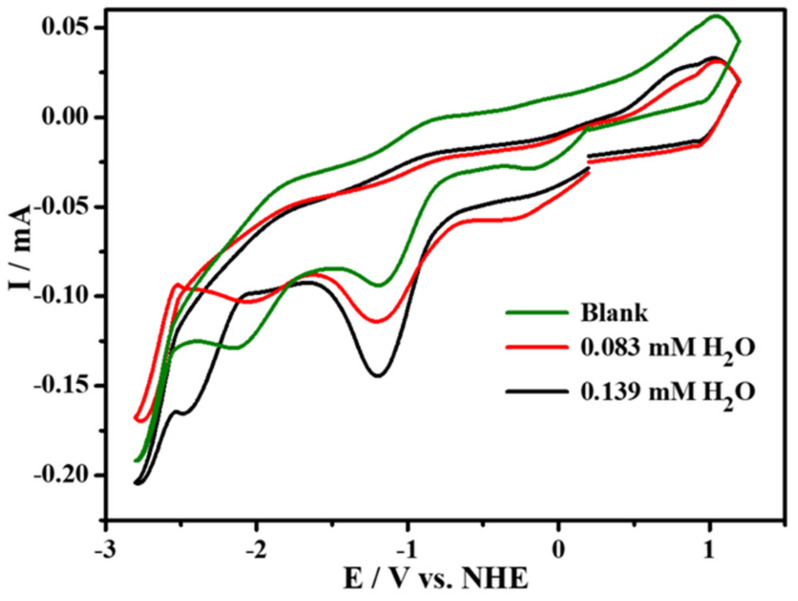
Cyclic voltammograms of the complex **1** containing the H_2_O system at different concentrations in the CO_2_ atmosphere. Scan rate: 100 mV s^−1^. Working electrode: glassy carbon. Counter-electrode: Pt wire. Reference electrode: Ag/AgCl.

**Figure 6 molecules-27-01399-f006:**
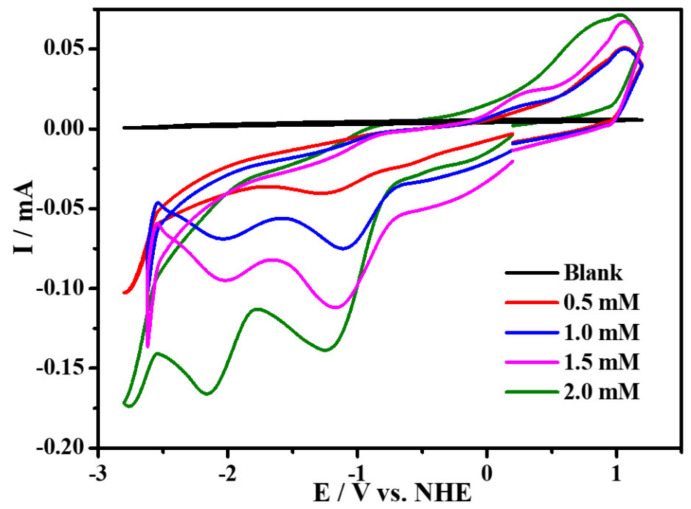
Cyclic voltammograms of the complex **1** at different concentrations in the CO_2_ atmosphere. Scan rate: 100 mV s^−1^.

**Figure 7 molecules-27-01399-f007:**
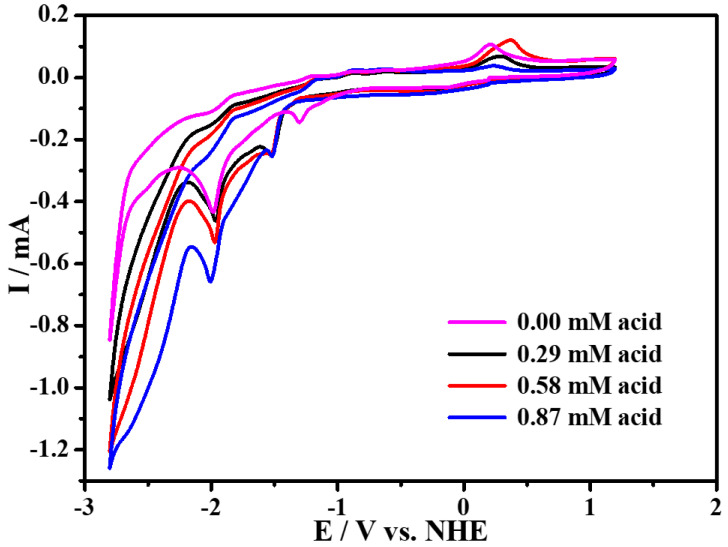
Cyclic voltammograms of complex **1** (2 mM) recorded in the absence (rose red trace) and in the presence of TsOH·H_2_O: 1 equiv (black trace), 2 equiv (red trace), and 3 equiv (blue trace) in CH_3_CN (0.1 M^n^Bu_4_NPF_6_) at a glassy carbon electrode and 100 mV s^−1^.

**Figure 8 molecules-27-01399-f008:**
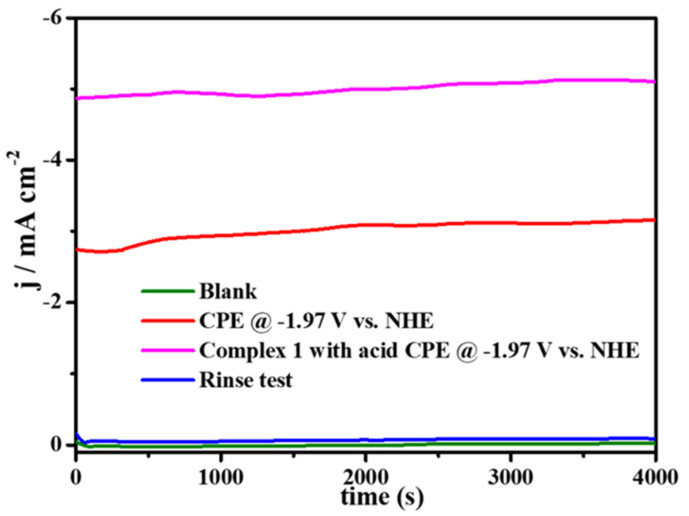
CPE of 2 mM complex **1** in CH_3_CN (0.1 M ^n^Bu_4_NPF_6_) (red) or in CH_3_CN (0.1 M ^n^Bu_4_NPF_6_) (rose red) solutions with 0.58 mM TsOH·H_2_O added; no complex **1** (green) under an atmosphere of Ar on the FTO working electrode; rinse test (blue).

## Data Availability

The data presented in this study are available on request from the corresponding author. The data are not publicly available due to this being part of federally funded research.

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
