# Peer review of "Electrocatalytic CO2 Reduction and H2 Evolution by a Copper (II) Complex with Redox-Active Ligand"

_molecules, 2022, doi:10.3390/molecules27041399_

Round 1
Reviewer 1 Report
- Figure 3a demonstrate cyclic voltammograms of 2 mM complex under 1 atm CO2. Author should clarify the plots is subtracted from the baseline or the peak at 1.15V can not represent the CO2R reaction.
- Please explain the purpose of the figure3b in the paper. It looks needless in the paper.
- Figure 3a 100mV/s line is not match with figure 4 blank line. Because they are under same condition, this huge difference imply a reproducible problem and will cause all the conclusions become unreliable.
- Same as comment 2. 100mV/s line in figure 3a, blank line in figure 4 and 2mM line in figure 5a should be same. However, they have a huge difference. Which means, these data looks like just one-time data and cannot reproduce. So, it makes me feel not confident to believe all the conclusions in this work. The author should explain it.
Reviewer 2 Report
1) When authors descibing metal complexes, the oxidation state must in all cases be indicated in brackets. This also applies to the complex 1 described in the article and fragments of the introduction, which lists examples from the literature (complexes of copper (II), cobalt (II) etc)
2) Page 2 lines 73-74
copper (II) complex [Cu (L1) 2NO3] NO3 (1) coordinated by this ligandIt is not a complex that can be coordinated through a ligand, but a metal ion.
3) The coordination polyhedron of copper in 1 must be analyzed using the SHAPE package or a similar program. Obviously, the distorted geometry is caused by the Jahn-Teller effect, but it may turn out that the elongated octahedron is not the most accurate description of the polyhedron.
See refs
M. Llunell, D. Casanova, J. Cirera, P. Alemany, S. Alvarez, SHAPE v. 2.1. Program for the stereochemical analysis of molecular fragments by means of continuous shape measures and associated tools, (2013) SHAPE v. 2.1. Program for the stereochemical analys.
Six vertex polyhedra: S. Alvarez, D. Avnir, M. Llunell, M. Pinsky. New J. Chem. 26, 996 (2002).
4) The picture with voltmperograms (figure 2a) does not allow you to see the features, the lines merged. Whenever possible, the data should be presented in a more descriptive form. This also applies to pictures 3-5. It is not clear why graph 4b similar to the existing ones 2b, 3b, 5b is not shown,
5) Is it possible to bring images of LUMO-1, LUMO-2 orbitals, since reduction is possible not only for the first stage?
6) The Y scale on graph 8 was chosen unsuccessfully, Ymax should be no more than -6. It is possible to bring pink and blue graphs on one image, and green on another, or use a graph with a partial scale break (between Y = -2.5 and ~ -0.1)
7) It is necessary to provide details of the DFT calculations!
Reviewer 3 Report
The paper describes the synthesis and characterisation of a copper complex bearing a benzimidazole-based ligand. The complex is studied to evaluate its performance for CO2 and H+ homogeneous electroreduction. Several typos and English grammar. E.g.: line 30 page 1 “have been being”, Line 47 page 2 “various reaction” were detected. Moreover, there are missing literature references for homogeneous copper catalysts already used in CO2 and H+ electroreduction.
Apart from these, I believe that this work needs further improvement and I consider that experiments are needed to clarify crucial points. Therefore, it is my believe that this paper is not suitable for Molecules.
Major concerns:
- The authors should clarify copper’s oxidation state by using, for example, EPR or Magnetometry.
- Under CO2 atmosphere, the current in all potential range drastically decreases when compared with the experiment under Ar. How do the authors explain this? If a catalytic behaviour is present, an increase in current is typically observed.
- The TOF should be calculated after CPE experiments (TON = mol product/ mol catalyst) to avoid overestimation of this value by using Eq. 1. The TON value should support the catalytic behaviour.
- Line 154 and 155 page 6: The authors state that the calculated TOF is comparable with other copper based homogeneous catalysts from the references 13, 26 and 27. However, only 26 reports a homogeneous catalyst but not for CO2 reduction. Please, compare with the proper references for the homogeneous copper complexes used in CO2 electroreduction. The same occurs in line 207 and 208 page 8: the authors state that the TON is comparable with the homogeneous catalysts in the references 11 and 32. Please provide the correct references for copper homogeneous catalysts used in H+ electroreduction.
- Why the authors performed the electrolysis using FTO electrode and the CVs with GCE? The authors should perform electrolysis with GCE and check if the behaviour is similar (same FE for CO and H2).
- DLS before and after electrolysis will check if solid state copper compounds can be formed.
Minor concerns:
- Figure 2 (b): the authors should check the linear relationship for all peaks.
- Cyclic voltammetry under Ar atmosphere for the ligand will help to prove the non-innocent redox behaviour of the ligand. In addition, I believe that from the observation of the LUMO and HOMO orbitals, is difficult to assume that the last reduction is metal-based (line 111 page 3) as there is a clear mixture of ligand and metal.
- Line 129-131 page 4: the authors state that all processes are controlled by diffusion, however the linear relationship was only calculated for the process at -1.15 V vs. NHE.
- Line 140 page 5: even with the addition of a proton source the current observed is less that the current observed under Ar.
- Provide the chromatograms (in supporting information) and more detail on the CO and H2 detection.
- Details of the theoretical calculations are missing.
Round 2
Reviewer 1 Report
I think the revised manuscript is suitable for publication in the Molecules
Author Response
Thank you very much for your kind suggestion, dear reviewer.
Reviewer 2 Report
The authors took into account the comments of the reviewers and provided the necessary data, made the figures more understandable and readable. I think that the article can be published in Molecules.
Author Response

(The authors gave the same response as above.)

Reviewer 3 Report
After considering the modifications made by the authors to improve the paper, I believe that is suitable for Molecules after the following minor revisions:
- In comment 3 “The TOF should be calculated after CPE experiments (TON = mol product/ mol catalyst) to avoid overestimation of this value by using Eq. 1. The TON value should support the catalytic behaviour.” The authors replied only with the calculation of this value for the HER reaction. I was wondering about the value for the CO2 If we are calling this system catalytic for CO2, these values should be reflected on the TON and TON values that are not overestimated.
- The response of the authors to comment 5 shows the CPE using GCE electrode. I understand that the authors gave some explanation about the benefits of using FTO to check complex stability. This should be clearly stated in the main text. In addition, the potential used in the CPE with GCE is not the same as the one used for the CPE using FTO (-1.15 V vs NHE). The main objective was to compare and check the similar behaviour between FTO and GCE, therefore the same potential should be used.
- Figure A5: please use the same convention to represent all cyclic voltammograms.
